# Novel Inhibitory Role of Fenofibric Acid by Targeting Cryptic Site on the RBD of SARS-CoV-2

**DOI:** 10.3390/biom13020359

**Published:** 2023-02-14

**Authors:** Jianxiang Huang, Kevin C. Chan, Ruhong Zhou

**Affiliations:** 1Institute of Quantitative Biology, College of Life Sciences, Zhejiang University, Hangzhou 310027, China; 2Shanghai Institute for Advanced Study, Zhejiang University, Shanghai 201203, China; 3The First Affiliated Hospital, School of Medicine, Zhejiang University, Hangzhou 310058, China; 4Department of Chemistry, Colombia University, New York, NY 10027, USA

**Keywords:** SARS-CoV-2, receptor-binding domain (RBD), fenofibric acid, drug repurposing

## Abstract

The emergence of the recent pandemic causing severe acute respiratory syndrome coronavirus 2 (SARS-CoV-2) has created an alarming situation worldwide. It also prompted extensive research on drug repurposing to find a potential treatment for SARS-CoV-2 infection. An active metabolite of the hyperlipidemic drug fenofibrate (also called fenofibric acid or FA) was found to destabilize the receptor-binding domain (RBD) of the viral spike protein and therefore inhibit its binding to human angiotensin-converting enzyme 2 (hACE2) receptor. Despite being considered as a potential drug candidate for SARS-CoV-2, FA’s inhibitory mechanism remains to be elucidated. We used molecular dynamics (MD) simulations to investigate the binding of FA to the RBD of the SARS-CoV-2 spike protein and revealed a potential cryptic FA binding site. Free energy calculations were performed for different FA-bound RBD complexes. The results suggest that the interaction of FA with the cryptic binding site of RBD alters the conformation of the binding loop of RBD and effectively reduces its binding affinity towards ACE2. Our study provides new insights for the design of SARS-CoV-2 inhibitors targeting cryptic sites on the RBD of SARS-CoV-2.

## 1. Introduction

The severe acute respiratory syndrome coronavirus (SARS-CoV-2) is the causative agent of the prolonged COVID-19 pandemic and remains widespread among the human population. To date, more than 600 million people have been infected and more than 6 million deaths have been recorded worldwide. Efforts have been devoted to vaccine development and effective interventions against COVID-19. Consequently, COVID-19 vaccines have been found to effectively reduce the spread of viral infection and associated morbidity and mortality [1,2,3,4,5]. However, advanced efforts in finding alternate treatments are still highly desired to cure COVID-19 patients with severe symptoms and reduce the death rate [6,7].

To date, the US Food and Drug Administration (FDA) has approved several small molecule antivirals for SARS-CoV-2, including remdesivir (VekluryTM), a ribonucleotide inhibitor of SARS-CoV-2 RNA-dependent RNA polymerase (RdRp) [8], baricitinib (OlumiantTM), a selective inhibitor of host proteins JAK1 and JAK2 [9], ritonavir-boosted nirmatrelvir (PaxlovidTM), and molnupiravir (LagevrioTM). Nirmatrelvir is a reversible covalent inhibitor of SARS-CoV-2 major protease (Mpro) [10], which is boosted by ritonavir, an HIV-1 protease inhibitor that allows nirmatrelvir to be active for a longer period by decreasing its cytochrome P450 3A-mediated metabolism. Molnupiravir, like remdesivir, targets SARS-CoV-2 RdRp; however, unlike remdesivir, which acts as a delayed chain terminator to halt viral RNA synthesis, molnupiravir acts as a mutagenizing agent during viral replication, causing dysfunctional virus copies [11].

Advanced research involving drug repurposing could be utilized for the potential identification of new drugs against COVID-19. Recently, numerous strategies of drug repurposing for COVID-19 have been summarized in the literature [12,13,14,15]. In the drug repurposing study by Hu et al. [16], they reported that antibiotics cefotaxime and cefuroxime have high binding affinities towards the spike receptor-binding domain (RBD) of SARS-CoV-2 through large-scale molecular chip screening, and further ensemble docking analysis suggests that cefotaxime and cefuroxime can influence the critical interface sites at the interface of RBD and ACE2 complex. A study by Davies et al. [17] shows that the metabolite of fenofibrate (fenofibric acid or FA) destabilized the RBD protein of SARS-CoV-2 and significantly reduced infection rates in vitro. The study suggested that FAs inhibit SARS-CoV-2 infections by inhibiting the binding of spike RBD to ACE2 receptor. According to the study, the inhibition process mainly involves the destabilization of the spike RBD [17]. However, the detailed inhibitory mechanism is yet to be fully explored. 

Previous evidence suggests that FA has a novel mechanism of action to interfere with ACE2-mediated binding and cellular entry of SARS-CoV-2. Herein, we hypothesized that the binding of FA to the spike RBD contributes to this inhibition process. In this study, we explored the potential binding mechanism of FA binding to the RBD of SARS-CoV-2 through molecular dynamics (MD) simulation. The MD simulation results showed that the FA induces a conformational change in the RBD by stabilizing a potential cryptic binding site around the T470-F490 loop. This cryptic binding site was then verified through molecular docking and MM/GBSA (Molecular Mechanics with Generalized Born and Surface-Area solvation) calculations. Furthermore, a comparative analysis of the RBD-ACE2 and FA-bound RBD-ACE2 was also performed. The results showed that the FA-bound RBD possesses a weaker binding affinity to ACE2, attributed to the reduced contacts at the interaction interface.

## 2. Systems and Methods

### 2.1. Cavity Search

CavityPlus [18,19] is a website that offers protein cavity detection and various functional analyses. CavityPlus employs CAVITY [20] to discover possible binding sites on the surface of a given protein structure using a structural geometry-based algorithm and then rank them based on ligandability and druggability scores using protein three-dimensional structural information as input. These possible binding sites may be investigated further with the help of three submodules: CavPharmer [21], CorrSite [22], and CovCys [23]. CavPharmer extracts pharmacophore characteristics within cavities using Pocket, a receptor-based pharmacophore modeling application. CorrSite uses motion correlation analysis between cavities to identify possible allosteric ligand-binding sites. CovCys discovers druggable cysteine residues. Overall, CavityPlus is a comprehensive platform for analyzing the features of protein binding cavities. CavityPlus webserver was employed for the detection of ligand-binding sites on the RBD of SARS-CoV-2 with default parameters. The three-dimensional protein model of RBD was downloaded from the protein data bank using the PDB code of 6VW1 [24] with the 2.68 Å X-ray resolution. The detected binding sites were graded using the metrics maximum pKd, DrugScore, and Druggability.

### 2.2. MD Simulation System and Setup 

Initially, the missing residues of the RBD protein structure were added using PDBfixer [25]. Similarly, the initial structure of fenofibrate was downloaded from the database ChemSpider [26]. Subsequently, the fenofibrate structure was edited using Avogadro software [27] to generate the initial FA structure (Appendix A) and was then subjected to geometrical optimization using MMFF94 force field [28].

Several parameters from the General Amber Force Field (GAFF) [29] were utilized to describe the FA. Based on the *pKa* value (4.0) of FA, deprotonation of the structure (carrying one negative charge) was performed under physiological conditions [30], while the partial charges for FA were derived by the AM1-BCC method [31]. The parameters were set for the protein according to the AMBER 99SB-ildn force field [32] and used in combination with the TIP3P explicit water model [33]. Following similar protocols used in our previous studies [34,35,36,37,38], addition of ions (Na^+^ and Cl^-^) was achieved to neutralize the system and yield an ionic concentration of 150 mM. All the MD simulations were performed using the GROMACS 2020.4 [39] simulation package whereas the temperature (T = 310 K) and pressure (*p* = 1 atm) were maintained using a stochastic velocity rescaling thermostat [40] and a Parrinello–Rahman barostat [41], respectively. The short-range electrostatic and van der Waals interactions were calculated at a cut-off distance of 1.2 nm, while long-range electrostatic interactions were treated via the particle mesh Ewald (PME) method [42]. The periodic boundary conditions were applied to each of the systems in all directions. Furthermore, the use of the LINCS algorithm enabled a standard integration time step of 2 fs [43]. Finally, 2000 ns MD simulations were performed for RBD and the four FA molecules. The MD simulations of RBD-ACE2 and FA-bound RBD-ACE2 were separately performed for 500 ns. The detailed information of the simulated systems was summarized in Appendix A. Similarly, all the simulation snapshots were rendered with VMD [44]. 

### 2.3. Molecular Docking and MM/GBSA Calculation

Molecular docking of FA to the RBD of SARS-CoV-2 was performed using AutoDock Vina [45]. The 2000 ns MD trajectory of RBD and four FA molecules were extracted every 2 ns to obtain a total of 1000 frames of RBD structures. The protein and ligand structures were converted into PDBQT format files using the AutoDock tools software [46]. Subsequently, a grid box of size 3 × 3 × 3 nm^3^ and an exhaustiveness value of 128 with other default parameters were adopted for the molecular docking calculations. The obtained molecular docking results were then ranked on the basis of predicted binding affinity. The complex structures selected from the top docking poses were then subjected to further MD simulations. Both the ligand–protein and protein–protein binding affinities were calculated by using the MM/GBSA method [47,48]. The MM/GBSA calculations were performed with the gmx_MMPBSA software [49]. Briefly, binding free energy (ΔG) was calculated by summing up the changes in electrostatic energies (ΔE^ele^), the van der Waals energies (ΔE^vdW^), the electrostatic solvation energy (ΔG^GB^, polar contribution), the nonpolar contribution (ΔG^SA^) between the solute and the continuum solvent, and conformational entropy (–TΔS) upon ligand binding [47,48].

## 3. Results and Discussion

### 3.1. Identifying Binding Sites on the RBD of SARS-CoV-2 Spike Protein 

The core of the SARS-CoV-2 RBD is a twisted five-stranded antiparallel sheet with short connecting helices and loops [50,51] and the receptor-binding motif (RBM) of RBD consists mostly of the upper loops, including the T470-F490 loop (shown in orange in Figure 1A). The potential binding sites on the RBD of SARS-CoV-2 spike protein were first explored by using CavityPlus [18]. As shown in Figure 1B, a total of six binding sites were predicted and each ranked based on DrugScore^14^. Cavity 1 had the highest DrugScore and largest vacant volume of 372.5 Å^3^ (Appendix A) and was therefore identified as the most favorable binding pocket for the FA molecule. Residues of the RBD protein surrounding the cavities are listed in Appendix A.

Next, we performed MD simulations of the RBD in the presence of four FA molecules randomly placed in solvent (Figure 2A). We also noticed that the T470-F490 loop of the RBD shows a relatively large conformational change during MD simulations. This conformation change was further quantified through root mean square deviation (RMSD) analysis (Figure 2B). The RMSD of the RBD was found to increase to ~0.6 nm upon binding of FA *I*. These conformational changes are illustrated as representative snapshots during the time evolutions of RMSD and shown in Figure 2C. To corroborate the observations, four more replicas of MD simulations were performed and the results of RMSD are shown in Appendix A. The RMSD of the RBD protein stays within 0.2–0.25 nm during the entire 2000 ns simulation time for all the four replicas, indicating an overall stable protein system in our simulations. The T470-F490 loop of the RBD protein of course shows more pronounced conformational changes, as expected (Appendix A). These additional simulations show very similar behaviors and trends with the one mentioned above (Figure 2), thus we stayed with that one as the representative for the rest of the analyses in this study. 

To detect the potential contact between the RBD and FAs, we calculated the time evolution of center-of-mass (COM) distances between the RBD and each of the four FAs (Appendix A). The COM distance result suggests FA *I* contact with RBD (with COM distances of ~4 nm). A closer examination of the simulation trajectory (Movie S1) and the time evolution of the COMs of the four FA molecules (Appendix A) suggested that FA *I* wandered around and formed many contacts with the RBD protein. On the other hand, Appendix A suggest that the other three FA molecules (i.e., FA *II*, *III,* and *IV*) bound to RBD within the first 50 ns and remained bound to RBD until at least ~1500 ns (with COM distances of ~4 nm). Furthermore, we calculated the contact probability for each FA molecule to the RBD (Appendix A). It can be seen from Appendix A that there is noticeable contact between FA *I* and the T470-F490 loop of the RBD protein. While FA *II* preferentially binds to cavity 6, both FA *III* and *IV* have a greater tendency to bind to cavity 1 (also see Figure 1 and Appendix A). Meanwhile, although the RMSD result of RBD shows a subtle conformational change of the protein at ~140 ns in comparison to the structure at 100 ns (Figure 2C), FA *I* remains in the solution and stays away from the T470-F490 loop of RBD from 100 ns to 140 ns (Appendix A). The FA *I* also forms contacts with RBD only after 400 ns of MD simulations. These close interactions between the FAs and the T470-F490 loop are displayed in the representative snapshots at 338, 440, and 1772 ns (Figure 2C). Meanwhile, the increased RMSD of the protein at these time points suggests that the conformational change of the T470-F490 loop of the RBD is potentially induced by the FA *I* molecule (Movie S1). 

### 3.2. Principal Component Analysis (PCA) of MD Simulation Trajectory and FTMap Analysis

Next, we explored the effects of FAs on the conformational change in the RBD of the spike protein. The time evolution of the protein structures is presented in Figure 3A which represents different structures colored on the basis of simulation time from red to white and blue. The T470-F490 loop region showed the largest conformational changes, while the rest of the protein remains quite rigid. Additionally, we also performed principal component analysis (PCA) analysis to reveal protein conformational changes along the MD trajectories. Two top PCs used for projection analysis of the MD trajectory allow the detection of conformational change (Figure 3B). Consistent with the findings of the RMSD result (Figure 2B), we observed three different trajectory projections (i.e., three clusters) in the PCA result and colored these as red (0−388 ns) to gray (389−1769 ns) and blue (1770−2000 ns), respectively. The relatively long difference between the early and the late stage of the trajectory confirms that the T470-F490 loop of RBD underwent large conformational changes. This result is also consistent with the anisotropic network model (ANM) [52] analysis of RBD, where the collective motions in the RBD were predicted using coarse-grained normal mode analysis (Appendix A). The ANM analysis also suggests that the largest mean fluctuation is located at the T470-F490 loop of RBD. Moreover, the T470-F490 loop fluctuations are also in line with the simulation results of the previous different studies [53,54,55]. 

Furthermore, we adopted a centroid-based clustering method to determine the representative structures of the three clusters. To begin with, centers of the three clusters were determined by averaging the two PCs and then the RBD structures with the minimum distances to the three cluster centers on the two-dimensional PCA were taken as the representative ones. Finally, the RBD protein structures at 70, 464 and 1996 ns were determined as representative ones for the three clusters (Appendix A). Interestingly, while no contact between FA *I* molecule with the T470-F490 loop of the RBD protein at 70 ns was observed and the RBD protein remained similar to its crystal structure, direct contacts between FA *I* molecule with the T470-F490 loop of the RBD protein were heavily observed around the 464 and 1996 ns clusters, suggesting the potentially important role of FA molecule in inducing the conformational change of RBD.

We observed the binding of FA *I* to the T470-F490 loop of RBD after 1807 ns of MD simulations (Figure 3C). This observation was compelling to explore whether this loop region could be a potential cryptic binding site [56]. By definition, cryptic binding sites are not formed in protein targets without a ligand and only become visible upon the binding of ligands [57]. Herein, we additionally performed FTMap analysis [58] for the RBD protein structure at 1807 ns to detect any potential cryptic binding site. FTMap utilizes small organic molecules as probes to scan for pockets on the surface of a target protein. Moreover, it also helps in exploring favorable positions, clustering conformations, and ranking based on average energies [58]. Here, two major consensus sites (CS) were detected by FTMap (Figure 3D). We found that CS1 was in place with Cavity 1 detected by CavityPlus (Figure 1B). CS2 was in close proximity to the T470-F490 loop, suggesting that the T470-F490 loop can indeed be a potential cryptic site for ligand binding. 

Note that FA consists of two hydrophobic benzene moieties and one carboxyl group (Appendix A). The simulated system of one RBD and four FA molecules is similar to the use of cosolvent method to detect cryptic binding sites [57,59], where a number of organic solvents, such as phenol were added to MD simulations of the targeted protein. Due to the hydrophobic features of the cryptic site [56], organic solvents with hydrophobic moieties are likely to induce the opening of cryptic sites during the MD simulations. Therefore, FA plays a similar role to the organic solvent adopted for the cosolvent MD simulations. This analogue explains the above observation that FA *I* induces the fluctuations of the T470-F490 loop and the exposure of the potential cryptic site that is closed in the apo crystal structure.

### 3.3. Molecular Docking of FA to RBD and Binding Affinities Calculated by the MM/GBSA Method 

Next, we also analyzed the binding of FA to the potential cryptic site in the loop region of RBD by performing molecular docking of FAs to the RBD structure (Figure 4A). The 2000 ns MD trajectory was then extracted every 2 ns to obtain a total of 1000 representative frames of the protein structures. Herein, the molecular docking of FA to the loop region of RBD was performed using AutoDock Vina [45]. The docking poses were then ranked by the predicted binding affinity (Appendix A). Interestingly, we observed that the docking poses with relatively high binding affinities are mainly centered towards RBD structures at ~400 ns and ~1800 ns and were found consistent with the time points of the RMSD increase (Figure 2B). This increase in RMSD also implies the formation of the potential cryptic site and may result in the higher binding affinities with the RBD structures at ~400 ns and ~1800 ns.

We selected the top 43 poses with predicted binding affinities lower than or equal to −7.2 kcal/mol and each performed 100 ns of MD simulations. The COM distances between RBD and FA were monitored for all the systems and only those with close and stable COM distances were subjected to the MM/GBSA calculations. The binding free energies of the top 17 systems calculated by the MM/GBSA are shown in Appendix A. The top four binding poses, denoted from pose *a* to *d*, are shown in Figure 4B and the corresponding binding affinities are shown in Figure 4C. Based on the related binding affinities of the top four complex structures, the MD simulations of these complexes were extended by another 400 ns. The binding affinities calculated by the MM/GBSA on the extended MD trajectories are shown in Figure 4C with the pose *d* showing the strongest ligand–protein binding affinity. 

Furthermore, we also performed the molecular docking of FA to the CS1 site predicted by FTMap. During this analysis, the binding pose with the highest binding affinity of −7.2 kcal/mol (Appendix A) was adopted for the 100 ns MD simulations. The results of the binding affinity of FA to the CS1 pocket by the MM/GBSA method are listed in Appendix A. We obtained a significantly weaker binding affinity of −2.40 kcal/mol in comparison with that from CS2. This comparison suggests that CS2 is more favorable for the binding of FA molecules.

### 3.4. Structural Analysis of the Complex with the Highest Binding Affinity

A detailed examination of the binding pose *d* suggested the two benzene ring moieties of FAs are surrounded by several hydrophobic residues (Figure 5A) which may contribute towards the hydrophobic interactions between FA and RBD. To comprehend the ligand–protein interactions, we further analyzed the extended 400 ns MD simulations of the pose *d* using the MD-IFP tool [60] (Figure 5B). Hydrophobic interactions and aromatic contacts were denoted by HY and AR, respectively, and HD denotes residues which serve as hydrogen bond donors. FAs were able to form stable hydrophobic contacts with residues of RBD including ILE472, TYR473, GLN474, THR478, and PHE486. Consistently, the hydrophobic interactions within the T470-F490 loop of RBD were well characterized in the previous studies [61,62] and these hydrophobic interactions in the T470-F490 loop are important for allosteric effects between residues at the ACE2-binding interface [63,64]. FAs also formed hydrogen bonds with residues including TYR473, GLN474, SER477, and THR478. These residues in the T470-F490 loop also play important roles in the binding to the ACE2 protein [61,62,63,65]. 

### 3.5. Potential Mechanism of FA Reducing the Complexation of RBD and Human ACE2

The binding of FA to the loop region of RBD can have significant influences on the binding of RBD to hACE2. In the crystal structure of RBD-ACE2 complex, the T470-F490 loop stably interacts with the ACE2 (Figure 6A), which involves residues pairs Y473-T27, F486-F28, F486-L79, F486-F28, Y489-Y83, Y489-F28, and A475-T27 (the upper right enlarged figure of Figure 6A). By contrast, the binding of FA disrupted the binding interface of RBD for ACE2 near the T470-F490 loop (the lower right enlarged figure of Figure 6A). As a result, the residue contacts between RBD and ACE2 were significantly reduced in the FA-bound RBD-ACE2 complex. 

We further investigated the energetic contributions caused by the binding of FA to the T470-F490 loop of RBD to RBD-ACE2 interactions. Herein, both the structures of RBD-ACE2 and FA-bound RBD-ACE2 complexes were subjected to 500 ns MD simulations. The protein–protein binding affinities calculated by the MM/GBSA are shown in Figure 6B and Table 1. The entropy terms were omitted due to the difficulty in the entropy calculation for the relatively large protein complex. The calculated binding free energies are termed as effective binding free energies as shown in a previous study by He et al. [66]. The effective free energy of FA-bound RBD-ACE2 was −33.32 kcal/mol which is ~15 kcal/mol weaker than that of RBD-ACE2 (−47.35 kcal/mol), demonstrating a significant abolishment in ACE-binding induced by FA. This difference can be attributed to the deviations in vdW (ΔE^vdW^) and electrostatic (ΔE^ele^) terms due to the restraint of loop residues from the binding interface by the bound FA. Our results are in good agreement with the previous computational study by He et al. [66], in which they studied the molecular mechanisms of human infection with SARS-CoV-2 and SARS-CoV by using MM/GBSA calculations and they reported a binding affinity of −50.43 kcal/mol for the apo SARS-CoV-2 RBD-ACE2 complex. By decomposing MM/GBSA results, we clearly showed that the weakened ACE2-binding caused by FA was mainly contributed by fewer van der Waals interactions between RBD and ACE2. 

## 4. Conclusions

We systematically investigated the interactions between FAs and RBD by utilizing molecular docking, MD simulations, and MM/GBSA calculations. We found that FA interacts with the T470-F490 loop of RBD which is a potential cryptic site and remains closed in the apo crystal structure. The relatively higher binding affinities confirmed that FA binds to the loop region instead of other predicted pockets. Further analysis revealed the significant role of hydrophobic interactions, hydrogen bonding, and electrostatic interactions involved in the binding of FA to the RBD protein. Moreover, the protein–protein binding affinities calculated by MM/GBSA illustrate that the FA-bound RBD possesses a weakened binding affinity towards ACE2 in comparison to apo RBD. These findings suggested that the binding of FA affects interactions between the RBD of SARS-CoV-2 spike and hACE2 receptor through stabilizing an alternative conformation of the T470-F490 loop of RBD. By revealing a cryptic site on SARS-CoV-2 RBD, we emphasized the potential of targeting cryptic pockets on SARS-CoV-2 for the development of novel drug candidates for COVID-19 [67,68]. Since the current study was conducted using computational approaches, it is highly desired to have these findings validated with wet-lab experimentation in future studies.

## Figures and Tables

**Figure 1 biomolecules-13-00359-f001:**
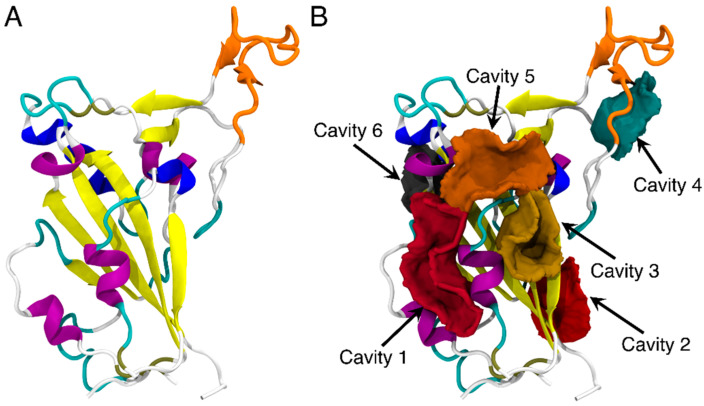
The structure of RBD of SARS-CoV-2 (**A**) and the predicted binding sites of RBD through CavityPlus analysis (**B**). The detected cavities are distinguished and represented in different colors. The RBD protein is shown in the cartoon representation. The T470-F490 loop is shown in orange, while the rest of the protein is colored by the secondary structures.

**Figure 2 biomolecules-13-00359-f002:**
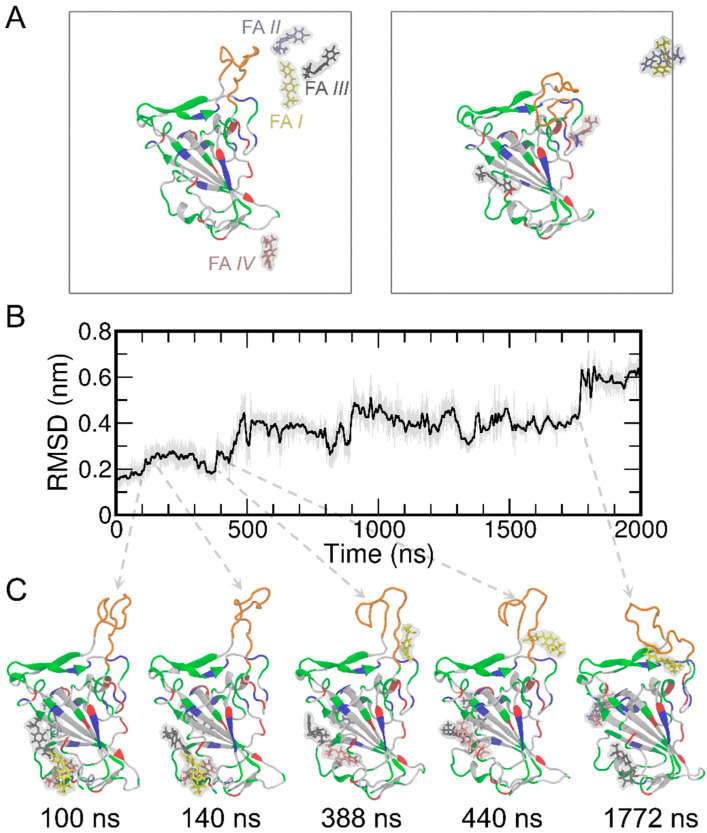
The initial and final structures of the MD simulations of RBD and four FA molecules (**A**) where the simulation boxes are outlined in gray. For clarity, the water molecules and ions were not displayed in the figure. The RBD protein is shown in the cartoon representation. The T470-F490 loop is shown in orange, while the rest of the protein are colored by the residue types (positively charged, negatively charged, polar, and nonpolar residues are shown in blue, red, green, and white, respectively). The FA molecules are shown in the stick representation and the surfaces are shown in transparent gray. The FA *I*, *II*, *III*, *IV* are shown in yellow, ice blue, gray, and pink, respectively. The RMSD results of the simulated RBD protein (**B**) and representative snapshots (**C**) are shown along the MD simulations.

**Figure 3 biomolecules-13-00359-f003:**
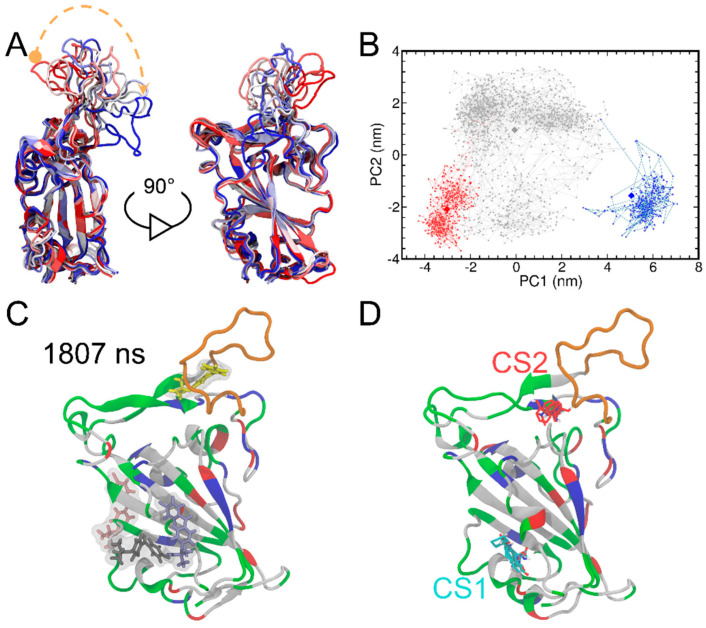
The superimposition of the RBD structures from MD simulations (**A**). The RBD protein is shown in the cartoon representation. The structures are colored on the basis of simulation time from red to white and blue. The curved dashed orange arrow highlights the time evolution of the loop fluctuations. The principal component analysis of the RBD structures from MD simulations is also shown (**B**). PCA results include graphs of PC1 vs. PC2 colored from red (0−388 ns) to gray (389−1769 ns) and blue (1770−2000 ns). The RBD structures at 70, 464 and 1996 ns (shown as diamonds) were determined as representative ones for the three clusters by using the centroid-based clustering method. The system snapshot at 1807 ns where FA *I* bind to the T470-F490 loop of RBD (**C**). The T470-F490 loop is shown in orange, while the rest of the protein is colored by the residue types (positively charged, negatively charged, polar, and nonpolar residues are shown in blue, red, green, and white, respectively). The FTMap analysis of the RBD protein structure at 1807 ns (**D**). The two major consensus sites (CS) (i.e., CS1 and CS2) indicated and small molecular probes in CS1 and CS2 are shown as cyan and red, respectively.

**Figure 4 biomolecules-13-00359-f004:**
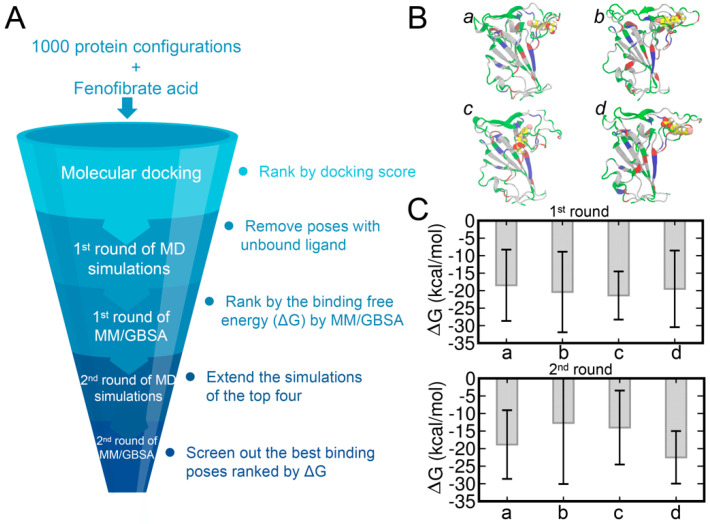
The workflow adopted for screening the best binding poses and results of MM/GBSA calculations (**A**). The top four binding poses (denoted as poses *a*, *b*, *c*, and *d*) (**B**). The protein structures were extracted from the time of 1774 ns (*a*), 1776 ns (*b*), 1786 ns (*c*), and 1788 ns (*d*). The FA molecules are shown in the sphere representation and the hydrogen, carbon, oxygen, and chloride atoms of FA are shown in white, yellow, red, and pink, respectively. Binding affinities of FA−RBD calculated by the MM/GBSA method (**C**).

**Figure 5 biomolecules-13-00359-f005:**
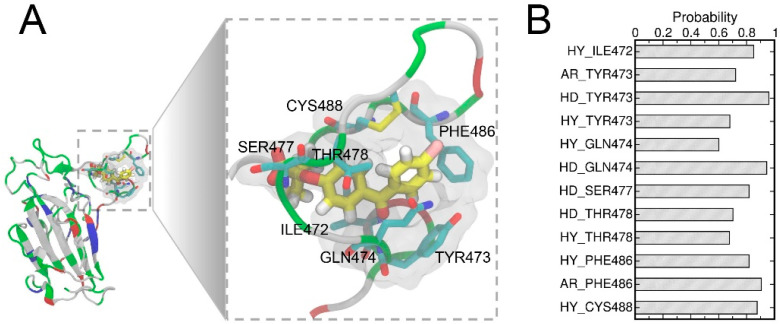
The binding pocket in a close view (**A**) showing the important labeled residues. The FA molecule is shown in the stick representation and the hydrogen, carbon, oxygen, and chloride atoms of FA are shown in white, yellow, red, and pink, respectively. The important protein–ligand interactions were analyzed using the MD-IFP tool [60], whereas HY and AR denote hydrophobic interactions and aromatic contacts with the protein residues, respectively, while HD denotes residues acting as hydrogen bond donors (**B**).

**Figure 6 biomolecules-13-00359-f006:**
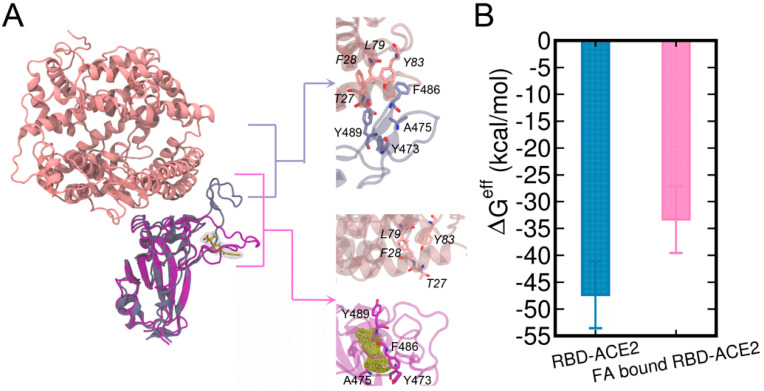
RBD-ACE2 complex and FA-bound RBD-ACE2 complex are aligned by the RBD (**A**). The structures of ACE2 and RBD are shown in pink and ice blue cartoons while the FA-bound RBD is shown in purple, respectively. FA molecule is shown in the yellow meshed surface. The effective binding free energies (ΔG^eff^) [66] calculated using the MM/GBSA method are also shown (**B**).

**Table 1 biomolecules-13-00359-t001:** The free energy results obtained from MM/GBSA calculations.

Complex	ΔE^ele^	ΔE^vdW^	ΔG^GB^	ΔG^SA^	ΔG^eff^ *
FA-bound RBD-ACE2	−535.22	−65.21	577.13	−9.03	−33.32
RBD-ACE2	−551.12	−89.79	607.62	−13.06	−47.35

*: ΔG^eff^: effective free energy calculated by the sum of ΔE^ele^, ΔE^vdW^, ΔG^GB^, and ΔG^SA^.

## Data Availability

The PDB structure was downloaded from https://www.rcsb.org/. The data presented in this study are available upon request from the corresponding author.

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
