# Peer review of "Novel Inhibitory Role of Fenofibric Acid by Targeting Cryptic Site on the RBD of SARS-CoV-2"

_biomolecules, 2023, doi:10.3390/biom13020359_

Round 1

Reviewer 1 Report

Overall, this MD study is exciting as it explores the processes by which fenofibrate (FA) drug binds to the SARS-CoV-2 Spike protein receptor binding domain (RBD). The authors demonstrated that FA binding to RBD shifted an ACE-interacting loop of RBD downward and exposing a cryptic site for FA to bind. As a result, the binding affinity of Spike protein to ACE2 is decreased. The analysis was well-planned. Before it is prepared for publishing, however, the following issues should be resolved:

Major:

1.     To corroborate the observations, please repeat the main MD simulation in triplicates.

2.     It's difficult to understand the explanation of section 3.1. The authors briefly describe each CavityPlus pocket before mentioning just one FA bind to cavity 1 after 2000 ns. The reader gets the impression that no other FAs were bound before the 2000. The authors then began to explain how other FAs also bind to the protein in the early stages of the MD simulation, which is very confusing given that they started out by stating that.

To make it easier to read, I believe the authors should first describe the structure of RBD, including which part binds to ACE2 and mention T470-F490 loop. Afterward, preclude your discussion on CavityPlus pockets before beginning to explain the simulation from beginning to end. Make sure to specify which pocket they bind when explaining the binding of FA during the explanation (to correlate with the Cavity Plus defined pocket). It would be fantastic to colour each FA in different colour so that readers can easily identify each FA.  

3.     I do agree with the authors that FA binding causes the shift of the T470-F490 loop to come down, but I'm not sure that this effect is solely the result of FA binding in cavity 1. Other FA do bind the cavity 6 in the figure, as well. I'm not sure if it's a result of both cavities working together or just pure activation through cavity 1. To confirm this, feel free to run the analysis again with just one molecule.

4.     Movie 1 is not included in the supplemental materials. I am relying on my observations looking at the figure only. 

Minor:

1.          Introduction --  more details on approved SARS-CoV-2 drugs that are currently known need to be added. Which protein is the drug's target, and what is the mechanism? It would be great to include RBD-targeting drugs, even if they are not yet approved.

2.        Discussion --  Please explained what was done by He et al. and what they shown … 

3.          Please provide X-ray resolution of structure 6vw1

Reviewer 2 Report

Fenofibric acid (FA) binds to RBD of spike protein and inhibits the interaction of RBD with hACE2. The authors have tried to decipher the inhibitory mechanism of FA using simulation studies and free energy calculations. Finally the authors have reported cryptic site around T470-F490 loop on RBD, where FA can interact and inhibit the interaction with hACE2.

Comments:

1. In line 115: Was there any distance criteria used for placing the FA molecules?

2. In Fig2- label/indicate the RBD loop that undergoes conformational change.

3. In line 117, the authors mention one FA binds to RBD and 3 remain unbound. However, in line 139, they mention that 3 FAs bind to RBD in first 50 ns.  Please clarify this ambiguity.

Again, in line 143, it is stated that FA I contacts RBD only after 400 ns. This is confusing, hence clarify in text and label, the 4 FAs in the Fig2 (panel A & C) for clarity.

4. The FA numbers should be labeled in Fig3C for clarity.

5. In the PCA analysis Fig3B, were there preferences in the FA molecules binding and localization on PCA map (red, grey & blue). If so, please show the structures of RBDs with the FAs bound, representing the above clusters.

6. These findings must be validated using wetlab experimentation, this should be elaborated in the conclusion section.

Reviewer 3 Report

In this work, Huang et al. applied multiple calculation methods, including docking, molecular dynamics simulations, MM/GBSA calculation, to perform a systematic investigation on the binding of FAs to RBD, which is the crucial recognition domain of the viral spike protein in SARS-CoV-2. The authors applied a series of elegant theoretical techniques in order to characterize the binding processes of FAs to RBD and address the molecular mechanisms of the binding. In detail, they did cavity analysis at the beginning, then the results were assessed by the long-time MD simulations, of which the results were further equipped to the pipeline of the molecular docking. Therefore, the study was rigorously done and the results were very convincing. I did go through all the procedures and read the details presented in the methods section, and did not see technique issues. In the end, the authors found that the T470-F490 loop, which plays important roles in the binding of RBD to ACE2 protein, is the most favorable target site for FA binding. The interactions of FA and RBD contribute to weakening the binding of RBD to ACE2 protein. Finding an effective drug for the treatment of COVID-19 has been a challenging task, the authors’ work provides an excellent theoretical evaluation on the potential role of FA on the inhibition of RBD binding to ACE2 receptor. Thus, the findings of the work are important, and I suggest publication on biomolecules journal.

I only have two comments (suggestions):

  1. It is hard to identify the exact position of T470-F490 in the RBD structures. The authors may consider to illustrate this loop in figure 1 or 2.
  2. The authors used CavityPlus to detect the potential binding sites of RBD. A concise description of the principle of the program would be helpful for the readers to understand their results.

Round 2

Reviewer 1 Report

The authors have performed all the suggested analysis. Well done, the manuscript is stronger and more convincing!! I think it's ready to be published. 

Reviewer 2 Report

Minor spell checks before printing might be required.

As of now the authors have  improved the manuscript after revisions.